# Extraction, Isolation, Characterization, and Biological Activity of Sulfated Polysaccharides Present in Ascidian Viscera *Microcosmus exasperatus*

**DOI:** 10.3390/ph16101401

**Published:** 2023-10-03

**Authors:** Ananda de Araujo Bento, Marianna Cardoso Maciel, Francisco Felipe Bezerra, Paulo Antônio de Souza Mourão, Mauro Sérgio Gonçalves Pavão, Mariana Paranhos Stelling

**Affiliations:** 1Federal Institute of Education, Science and Technology of Rio de Janeiro, Rio de Janeiro 20271-110, Brazil; nandah_araujo@hotmail.com (A.d.A.B.); marianna.maciel.etm2018.2@gmail.com (M.C.M.); 2Medical Biochemistry Institute, Federal University of Rio de Janeiro, Rio de Janeiro 21941-971, Brazil; felipebezerra_ipu@hotmail.com (F.F.B.); pmourao@hucff.ufrj.br (P.A.d.S.M.); mpavao@hucff.ufrj.br (M.S.G.P.)

**Keywords:** *Microcosmus exasperatus*, ascidians, sulfated glycosaminoglycans, cancer

## Abstract

Ascidians are marine invertebrates that synthesize sulfated glycosaminoglycans (GAGs) within their viscera. Ascidian GAGs are considered analogues of mammalian GAGs and possess great potential as bioactive compounds, presenting antitumoral and anticoagulant activity. Due to its worldwide occurrence and, therefore, being a suitable organism for large-scale mariculture in many marine environments, our main objectives are to study *Microcosmus exasperatus* GAGs regarding composition, structure, and biological activity. We also aim to develop efficient protocols for sulfated polysaccharides extraction and purification for large-scale production and clinical applications. GAGs derived from *M. exasperatus* viscera were extracted by proteolytic digestion, purified by ion-exchange liquid chromatography, and characterized by agarose gel electrophoresis and enzymatic treatments. Anticoagulant activity was evaluated by APTT assays. Antitumoral activity was assessed in an in vitro model of tumor cell culture using MTT, clonogenic, and wound healing assays, respectively. Our results show that *M. exasperatus* presents three distinct polysaccharides; among them, two were identified: a dermatan sulfate and a fucosylated dermatan sulfate. Antitumoral activity was confirmed for the total polysaccharides (TP). While short-term incubation does not affect tumor cell viability at low concentrations, long-term TP incubation decreases LLC tumor cell growth/proliferation at different concentrations. In addition, TP decreased tumor cell migration at different concentrations. In conclusion, we state that *M. exasperatus* presents great potential as an alternative GAG source, producing compounds with antitumoral properties at low concentrations that do not possess anticoagulant activity and do not enhance other aspects of malignancy, such as tumor cell migration. Our perspectives are to apply these molecules in future preclinical studies for cancer treatment as antitumoral agents to be combined with current treatments to potentiate therapeutic efficacy.

## 1. Introduction

Ascidians are marine invertebrates that comprise more than 2000 species [1,2,3,4,5,6,7]. They are classified into three orders: Aplousobranchia, Stolidobranchia, and Phlebobranchia [3,8,9,10,11,12,13]. Ascidians are filter feeders and can be found solitary or in colonies [14,15]. As hermaphrodite animals, they have developed remarkable reproductive strategies, combining sexual and asexual modes of reproduction, thus allowing rapid expansion of the individual population [3,16,17,18,19,20].

*Microcosmus exasperatus* (Stolidobranchia; Pyuridae) is an understudied species of ascidians found on the coast of Rio de Janeiro, Brazil. Nevertheless, this species has a worldwide distribution and is very common in tropical and subtropical waters. As such, *M. exasperatus* has been described as a marine biological indicator for monitoring stress and environmental pollution [21,22,23,24,25,26].

Studies have shown that *M. exasperatus* tunic is predominantly composed of sulfated polysaccharides, more specifically, high-molecular-weight sulfated L-galactans [27,28]. However, the types, structures, and biological activity of glycosaminoglycans (GAGs) present in the *M. exasperatus* viscera have not been evaluated until now.

Many species of ascidians, such as *Styela plicata*, *Phallusia nigra*, and *Ciona intestinalis*, are known and studied for synthesizing sulfated GAGs in their viscera and having other types of sulfated polysaccharides in their tunic, such as galactans and fucans [29,30]. These ascidian-derived GAGs are considered analogous to mammalian GAGs, playing essential roles in different biological processes such as tissue integrity, fertilization, infection, inflammation, cell growth, and others [31,32,33]. Therefore, ascidians are seen today as a promising natural source for the production of polysaccharides with antitumoral, anticoagulant, antimicrobial, antioxidant, neuritogenic, neuroprotective, and anti-inflammatory therapeutic properties [34,35,36,37,38].

Mammalian heparin is a sulfated GAG that has been widely used in the clinic as an effective anticoagulant and antithrombotic molecule. This molecule has also been studied as a promising alternative for cancer treatment and tumor progression [39,40]. Nevertheless, the occurrence of molecules such as heparin is not restricted to mammals [41], and the search for analogous compounds derived from marine invertebrates, such as ascidians, with similar biological activities but possibly devoid of unwanted side effects, offers a promising alternative in this area [42].

The ascidian *Styela plicata* has been described to synthesize low-anticoagulant heparin when compared to mammalian heparin. However, *S. plicata* heparin activates heparin cofactor II with approximately the same potency as vertebrate heparin. Studies comparing hemorrhagic effects between marine invertebrates and mammalian heparin revealed that ascidian heparin did not increase blood loss when compared to the control, while mammalian heparin increased blood loss by almost two times [35,41].

In addition to heparin, dermatan sulfates have also been described in ascidians. Interestingly, the ascidian DS sulfation pattern is order-specific, whereas Stolidobranchia species DS are mainly composed of IdoA2S-GalNAc4S, while Phlebobranchia species DS are primarily IdoA2S-GalNAc6S. The difference in the N-acetyl-galactosamine sulfation position of different ascidian dermatan sulfates confers different anticoagulant activities. IdoA2S-GalNAc4S DS is notably anticoagulant, while IdoA2S-GalNAc6S DS does not present detectable anticoagulant activity. The difference in the anticoagulant activity of these dermatan sulfates suggests that binding of GAGs to heparin cofactor II requires a specific sulfate pattern composed of enriched sequences of [α-L-IdoA(2SO4)-1→3β-D-GalNac(4SO4)] [35,41,43].

As previously mentioned, due to the clinical potential of marine invertebrates-derived GAGs and since they are analogous to mammalian heparin, it is important to establish a specific protocol for the collection, isolation, extraction, and purification of sulfated polysaccharides present in these animals’ viscera, specifically for this ascidian species. Unveiling the identity of *Microcosmus exasperatus* sulfated GAGs increases the library of bioactive compounds of marine origin. It is also important to evaluate if these molecules maintain their bioactivity after extraction and purification processes to fully explore their possible clinical applications.

## 2. Results

### 2.1. Morphology of the Ascidian Microcosmus Exasperatus

*M. exasperatus* ascidians present a densely inhabited tunic (Figure 1A), while the viscera presents an orange pigment (Figure 1C) that can be observed after dissection; finally, an intense purple color can be observed in the mantle (Figure 1B).

### 2.2. Microcosmus Exasperatus Viscera Presents a Dermatan Sulfate and a Fucosylated-Dermatan Sulfate

*M. exasperatus* total polysaccharides (TP) was analyzed regarding polysaccharide pattern through agarose gel electrophoresis. Three different bands can be observed in this analysis, and by comparison with commercially available standard GAGs, we identified a band migrating with standard dermatan sulfate (arrow A), another band migrating below standard DS and above standard heparin (arrow B), and, finally, a third band migrating with standard heparin (arrow C) (Figure 2A).

Next, TP was fractionated through ion exchange liquid chromatography on a DEAE cationic column. A stepwise gradient allowed the separation of two peaks, named sulfated polysaccharides 1 (SP1) and 2 (SP2) (Figure 2B). Interestingly, it was possible to observe that the polysaccharides produced specifically by this ascidian species are shifted from the cationic column at lower NaCl concentrations when compared to GAGs from other ascidian species [9,29,44], an indication that they present a lower negative charge density and, therefore, a lower sulfation degree.

SP1 and SP2 were separately characterized by agarose gel electrophoresis, and it was observed that SP1 comprises one isolated band migrating with standard heparin, while SP2 presents two bands (Figure 2C). The identity of the polysaccharides was further investigated through specific enzymatic treatments of the SP1 and SP2 fractions with a mixture of heparinases I and II, chondroitinases AC and ABC, and deamination by HNO_2_. As shown in Figure 2D, SP1 is resistant to all enzymatic treatments and HNO_2_ deamination. On the other hand, SP2 was degraded by chondroitinase ABC but not by chondroitinase AC, confirming its identity as a dermatan sulfate (Figure 2E).

In order to investigate SP1 identity, we applied nuclear magnetic resonance (NMR) spectroscopy (Figure 3). In Panel A of Figure 3, the comparison between the 1D and ^1^H NMR spectra from SP1 and the fucosylated-chondroitin sulfate (FUCCS) from *Ludwigothurea grisea* is presented in blue and red, respectively. Important regions such as anomeric and ring atoms are highlighted. The signals in the anomeric region were assigned as being from iduronic acid and related to fucose. We can also confirm the presence of hydrogen atoms bonded to carbon 6 in the CH_3_ region (marked as F6). The signal profile is similar to that found in FUCCS, due to their polymer nature.

Next, to overcome 1D ^1^H NMR overlapping signals, we performed a 2D ^1^H-^13^C HSQC experiment, shown in Figure 3B. We highlighted three important regions in the spectrum. Spectrum was assigned based on literature data (Table 1). We found characteristic signals of dermatan sulfate with iduronic acid (I) units, 4-sulfated GalNAc (A), 6-sulfated GalNAc (A’), and glucuronic acid (U). In addition to these signals, we also found signals related to fucose (F). These results point to a dermatan sulfate polymer with fucose substitutions. The combination of NMR data with SP1 resistance to chondroitinases treatment reinforces its identity as a fucosylated-dermatan sulfate.

We, therefore, conclude that *M. exasperatus* fraction SP1 presents a fucosylated-dermatan sulfate, and SP2 fraction presents a combination of a metachromatic uronic acid containing polysaccharide and a dermatan sulfate.

### 2.3. Microcosmus Exasperatus-Derived Dermatan Sulfate-Containing Fraction (SP2) Presents Anticoagulant Activity

In order to assess *M. exasperatus* GAGs anticoagulant potential, we investigated the SP1 and SP2 fractions’ performance in an activated partial thromboplastin time (APTT) assay. It was seen that the dermatan sulfate-containing fraction (SP2) presents mild anticoagulant activity, approximately one hundred times lower than the sixth International Standard for Unfractionated Heparin (Figure 4). On the other hand, the fucosylated dermatan sulfate (SP1) fraction did not present anticoagulant activity at the tested concentrations.

### 2.4. Microcosmus Exasperatus Total Polysaccharides (TP) Presents Antitumor Activity

Next, we investigated if *M. exasperatus* TP would present antitumor activity. Sulfated GAGs have been described as modulators of cell behavior, and such activity may vary from higher to lower concentrations. Therefore, we investigated the effects of *M. exasperatus* TP on tumor cell viability, colony growth, and migration using MTT, clonogenic, and wound healing assays, respectively. Figure 5 shows the cell viability and colony growth of the mouse Lewis lung carcinoma cell line (LLC). MTT assays were considered a short-term treatment, where cells were incubated with *M. exasperatus* TP for 24 h, while the clonogenic assay was considered a long-term treatment, where cells were incubated with TP for 72 h and colony growth was monitored for an additional 7 days.

We found that *M. exasperatus* TP presents different effects on LLC cell viability from lower (at 500 ng/mL) to higher (50 µg/mL) concentrations. The short-term incubation evaluated with the MTT assay revealed that the higher concentration seems to enhance cell viability. On the other hand, the long-term incubation evaluated using a clonogenic growth assay revealed that the lower and medium concentrations negatively affect colony growth, while the higher concentrations enhance colony growth.

Finally, we also investigated the effects of TP on LLC cell migration using the wound healing assay (Figure 6). Cells were monitored regarding wound closure activity, and, interestingly, *M. exasperatus* TP reduced LLC cell migration at 500 ng/mL and 50 µg/m. The combination of the acquired data indicates that lower TP concentrations, such as 500 ng/mL, should be considered for antitumor activity. The lowest concentration tested did not affect cell viability in short-term incubation; however, it was able to reduce cell growth and migratory activity in long-term incubation.

## 3. Discussion

Many ascidian species have already had their GAGs and associated biological activities described in the literature [35]. However, each species is a unique organism, presenting diversity in the variety, structure, and biological activity of these molecules. Therefore, it is very important to study new species and uncover the diversity of GAGs and sulfated polysaccharides to be considered as mammalian analogues and the specific biological activities presented by them. Therefore, a commonly found, but understudied, ascidian species identified as *Microcosmus exasperatus* was investigated as a promising candidate for the sustainable, large-scale production of mammalian heparin analogues. 

Ascidians were collected at different periods of the year to exclude possible seasonality in the glycan production of the species according to reproduction cycles. Such cycles occur in warmer temperatures (summer); however, when temperature peaks, water layer stratification occurs and, consequently, the depletion of food for ascidians. In contrast, when seawater cools (autumn), there is a decrease in the thermocline, causing the resurgence of deep-sea-rich food sources [45].

The first collection was carried out during the summer, and it was possible to observe a great occurrence of *M. exasperatus* specimens in Praia Vermelha, Rio de Janeiro. We obtained a greater number of specimens collected and, consequently, a higher amount in dry weight (81.98 g). The second collection, carried out in the winter period, also showed a high occurrence of these specimens, where 61.87 g of dry weight was obtained, but even so, lower than in the summer period. On the other hand, in the third collection during autumn, it was possible to observe a more discrete occurrence of these specimens in the same place, and, therefore, we obtained the lowest value in dry weight (18.86 g). Thus, our hypothesis is that *M. exasperatus* reproduces mainly in the summer, as observed by the abundance of young and adult specimens found during these collections, also corroborating with what is described in the literature regarding the short period necessary for ascidians development, with the larval stage lasting only a few minutes or, at most, a few hours. The differentiation of an adult individual with functional organs occurs in a few days to a few weeks [46,47,48]. However, it was also possible to observe a high occurrence of *M. exasperatus* specimens during the winter, probably due to the narrow temperature window typical of tropical regions. 

It is also important to emphasize that no apparent difference was seen in the number of organisms that inhabit the tunic of these ascidians in the three collections carried out, suggesting that the ecological relationship between epibiont organisms and ascidians is always present, regardless of seasonality, and may serve as a kind of camouflage or barrier to predators throughout the non-larval phase of ascidians [45,49].

After specimens’ collection, viscera dissection, and sulfated polysaccharide extraction were performed, a solution containing total polysaccharides, called TP, was obtained. Through agarose gel electrophoresis, it was possible to observe three types of uronic acid-containing sulfated polysaccharides present in this TP. Next, we fractionated *M. exasperatus* TP by ion exchange liquid chromatography. We observed that it was possible to fractionate these sulfated polysaccharides into two distinct fractions, called SP1 and SP2. Furthermore, analyzing the resulting chromatogram, we observed that these sulfated polysaccharides are eluted at lower NaCl concentrations when compared to sulfated polysaccharides from other ascidian species [9,35,43]. This data suggests that *M. exasperatus* sulfated polysaccharides present a lower negative charge density and, consequently, a lower sulfation degree. 

Identification of *M. exasperatus* GAGs species revealed the presence of a dermatan sulfate, a fucosylated dermatan sulfate, and an uronic acid-containing polysaccharide resistant to heparinases, chondroitinases, and HNO_2_ deamination. Interestingly, this is the first description of an ascidian-derived fucosylated dermatan sulfate. 

We have also analyzed the anticoagulant activity of the SP1 and SP2 fractions and observed that these fractions present mild (SP2) to absent (SP1) anticoagulant activity. As already described in the literature, the anticoagulant activity of dermatan sulfates from ascidians is strongly related to the GAG sulfation pattern. IdoA2S-GalNAc4S dermatan sulfates present significant anticoagulant activity, while IdoA2S-GalNAc6S dermatan sulfates do not present such activity [9,35,41,43]. Additionally, these sulfation patterns are order-specific, whereas *M. exasperatus*, a member of the Stolidobranchia order, should produce an anticoagulant dermatan sulfate. Therefore, *M. exasperatus* dermatan sulfate mild anticoagulant activity possibly results from the combination of a low degree of sulfation arranged as IdoA2S-GalNAc4S. The anticoagulant activity of the fucosylated dermatan sulfate (SP1 fraction) was also evaluated, and it was seen that this sulfated polysaccharide does not have anticoagulant activity at the tested concentrations. The absence of anticoagulant activity found in *M. exasperatus* fucosylated-dermatan sulfate was unexpected due to the description of fucosylated-chondroitin sulfate as having high anticoagulant activity [50]. 

Considering the growing interest in combination therapy to enhance cancer treatment efficacy, we sought to investigate if *M. exasperatus* polysaccharides would be a suitable alternative to the use of mammal heparin. *M. exasperatus* may be considered a good alternative to mammals because, as a marine invertebrate, its large-scale production could be directed towards a more sustainable approach, particularly requiring less dry land and freshwater to occur. Additionally, the evolutionary distance between ascidians and mammals adds to the safety against cross-species contamination; therefore, we consider *M. exasperatus* a particularly interesting organism to be considered for antitumor polysaccharide production.

The antitumor activity was evaluated using three different approaches in a cellular model using a mouse Lewis lung carcinoma cell line (LLC). We found that *M. exasperatus* total polysaccharides exerts different effects on tumor cells from short- to long-term treatments, as well as from low (0.5 µg/mL) to high (50 µg/mL) concentrations. We found that the high-TP dose acted as a tumor cell stimulant, enhancing viability and colony growth, while cell migration was reduced compared to controls. The low-TP dose, on the other hand, presented antitumor effects regarding colony growth and cell migration. While the short-term treatment did not present differences from the control condition.

In summary, our data points towards the use of long-term, very low doses of *M. exasperatus* TP, ideally well below anticoagulant concentration, when considering combination therapy. In addition, *M. exasperatus* TP extraction and possibly fractionation may easily be adapted to a large-scale regimen, as all steps may be executed in bulk. Finally, considering that this ascidian species can be found in many parts of the world, we highlight *M. exasperatus* as a promising organism for the extraction of unique sulfated polysaccharides, dermatan sulfate, and fucosylated-dermatan sulfate, with therapeutic properties as modulators of cell behavior. Marine invertebrates’ glycans represent a relevant and growing area of research, both in the clinical and environmental aspects of science, and further investigations in this area should be stimulated. 

## 4. Materials and Methods

### 4.1. Specimens Collection and Dissection

*Microcosmus exasperatus* ascidians species were collected (SISBIO permit: 66457-1) by free diving in Praia Vermelha, Rio de Janeiro, placed in collection buckets with seawater, and transported to the laboratory. At the laboratory, the viscera were dissected using a scalpel and scissors, cutting the tunic for viscera removal. Viscera were fixed in 100% ethanol, while tunics were frozen (for use in other projects) or discarded. 

### 4.2. GAGs Extraction from the Microcosmus Exasperatus Viscera

After *M. exasperatus* viscera dissection, GAG extraction was performed as described in [43] to obtain a final solution containing only the sulfated polysaccharides (named total polysaccharides—TP) and then perform the fractioning. 

#### 4.2.1. Delipidation and Depigmentation

To ensure that the extracted material was free of any fatty tissue and pigments, viscera were incubated in 92.8% ethanol at room temperature. Ethanol was replaced daily for a total of five days for the efficient removal of lipids and pigments. Next, the viscera were placed to dry in an incubator at 60 °C for 24 h. Once dry, the material was weighted.

#### 4.2.2. Proteolytic Digestion

The material was digested with papain in order to release polysaccharides from the tissue. The dry material was rehydrated for 2 h in a digestion buffer (100 mM sodium acetate; 5 mM EDTA; 5 mM cysteine; pH 5.0) at 5 mL per gram of dry material. Digestion was started by adding papain at a final concentration of 0.5 mg/mL and incubated at 60 °C for 24 h. After that, the material was mechanically homogenized using a spatula, additional papain was added, and samples were incubated at 60 °C. This process was repeated three times in order to complete the homogenization of the solid samples. The final digested suspension was centrifuged for 30 min at 3200 rpm, then the supernatant (the solubilized fraction containing the free polysaccharides) was collected, and the pellet (the undigested fraction) was discarded. 

#### 4.2.3. GAGs Precipitation

Polysaccharide precipitation in the supernatant after proteolytic digestion was performed by adding cetylpyridinium chloride (CPC) until reaching a final concentration of 0.5% *m*/*v*. After CPC addition, the final mixture was homogenized and incubated at room temperature for 24 h. Subsequently, the suspension was centrifuged at 3200 rpm for 30 min. The supernatant was discarded, and the pellet was completely dissolved in a 2 M sodium chloride (NaCl) solution containing 15% ethanol. After that, the polysaccharides free of CPC were precipitated by adding ethanol to 70% of the solution and incubating at −20 °C for 24 h. Next, the suspension was centrifuged for 30 min (3200 rpm), the supernatant was discarded, the pellet was further rinsed in 80% ethanol, centrifuged again for 30 min (3200 rpm), and the resulting pellet was air-dried. The pelleted TP was rehydrated in DNAse buffer (20 mM Tris-HCl pH 7.5; 5 mM NaCl; 3 mM MgCl_2_; 5 mM CaCl_2_) (1:1) and treated with 510 U Kunitz/mL of DNAse (Sigma, St. Louis, MO, USA) at 37 °C for 24 h. 

### 4.3. Sulfated Polysaccharides Electrophoretic Profile Analysis

After obtaining the TP, an agarose gel electrophoresis was performed to identify the polysaccharide electrophoretic profile found in *M. exasperatus* viscera, based on the comparison to commercially available GAG standards: porcine heparin, shark chondroitin sulfate, and mammalian dermatan sulfate. Samples were applied to a 0.5% agarose gel in 1.3-diaminopropane/50 mM acetic acid buffer, pH 9.0, and ran for approximately 2 h at 100 V. Polysaccharides were precipitated by gel immersion in a cetyltrimethylammonium bromide (cetavlon 0.1%) solution for 24 h. Next, the gel was dried under lamp heat (Ourolux 250 watts infrared), and the resulting dried gel sheet was stained with 0.1% toluidine blue in acetic acid, ethanol, and water (0.1:5:5, *v*/*v*) solution. 

### 4.4. Sulfated Polysaccharides Purification and Fractionation by Ion Exchange Liquid Chromatography

TP was fractionated by ion-exchange liquid chromatography. Initially, 200 µg of material were applied to a diethylaminoethyl cellulose anion exchange column (DEAE sepharose) (GE Healthcare) coupled to the FPLC (fast protein liquid chromatography) Äkta Prime equipment (Amersham Biosciences, Amersham, UK, SN: 1,102,380), equilibrated in 0.02 M Tris-HCl buffer, pH 8.0. The GAGs were eluted from the column in a linear NaCl gradient (0–3 M) using a 0.02 M Tris-HCl buffer containing 3 M NaCl, pH 8.0, at a flow rate of 3 mL per minute, with fractions of 3 mL per tube being collected. Next, conductivity was measured to assess the NaCl concentration. Thereby, 240, 370, and 700 mM NaCl concentrations were determined as concentrations used in a stepwise NaCl gradient for bulk sulfated polysaccharide fractionation. The sulfated polysaccharide elution pattern was assessed by metachromasia properties using 100 µL of DMB and 20 µL of sample in a 96-well plate. Then, read in a Versamax spectrophotometer with a microplate reader (Molecular Devices, San Jose, CA, USA) at 525 nm. 

Finally, the fractions containing metachromatic material correlated to the peaks in the graph (SP1 and SP2) were grouped and precipitated in 70% ethanol at −20 °C for 24 h. The suspension was centrifuged (3200 rpm for 30 min), the pellet was rinsed in 80% ethanol, centrifuged again (3200 rpm for 30 min), and let air dry. 

### 4.5. Lyases Specific Degradation and Nitrous Acid Deamination

For the sulfated polysaccharide identification, enzymatic treatments were performed on SP1 and SP2 with chondroitinases AC and ABC (Seikagaku, Japan), a mixture containing heparinases I and II, and deamination with nitrous acid (HNO_2_) to investigate the identity of the sulfated polysaccharide species present in the samples. Each sample was incubated for 24 h with 0.05 units of chondroitinases AC and ABC and 0.2 units of heparinases I and II in 2× digestion buffer (100 mM Tris-HCl, 30 mM sodium acetate, 1.0 mM EDTA, pH 8.0) for chondroitinases and (40 mM Tris-HCl, 100 mM NaCl, 8 mM CaCl_2_, pH 7.5) for heparinases at 37 °C, using sample, enzyme and 2× buffer ratio of 1:1:1. After 24 h, a new addition of each enzyme was made. 

Deamination was also performed, and the nitrous acid was produced from the reaction of sulfuric acid (H2SO_4_) at a final concentration of 0.25 M with sodium nitrite (NaNO_2_) at a concentration of 0.25 M for 20 min. Afterward, the samples were mixed with the same volume of nitrous acid (HNO_2_) (1:1, *v*/*v*) and incubated at room temperature for one hour.

Enzymatic degradation and deamination were evaluated by agarose gel electrophoresis for the comparison of intact and digested material, in addition to commercial standards for reference.

### 4.6. Uronic Acid Analysis

The polysaccharide concentration contained in the TP solution, SP1 and SP2, was estimated by its uronic acid content through the carbazole method [51]. 

Briefly, the method consists of incubating 10 μL of the polysaccharide solution with 190 μL of distilled water and 1 mL of sulfuric acid with 0.1% borate, homogenizing, and incubating for 12 min at 100 °C. After cooling, 40 μL of carbazole (0.2% in ethanol) are added to the solution, which is again homogenized and incubated at 100 °C for 10 min, then cooled once more, and the absorbance measured at 525 nm. The sample concentration was estimated from a glucuronolactone standard curve. 

### 4.7. Nuclear Magnetic Resonance (NMR) Analyses

Nuclear magnetic resonance (NMR, 1D, and 2D spectra) was performed for the SP1 fraction (2 mg) using a triple resonance probe (900 MHz Bruker, Karlsruhe, Germany). ^1^H NMR 1D spectra were recorded using 32 scans with a 1.5-s inter-scan delay at 35 °C. ^1^H/^13^C HSQC (heteronuclear single quantum coherence) spectrum using time proportion phase incrementation (TPPI) for quadrature detection in the indirect dimension. A total of 64 scans and 1048 × 512 points were recorded. 

### 4.8. Activated Partial Thromboplastin Time (APTT)

Anticoagulant activity was measured by the activated partial thromboplastin time (APTT) assay, as described by Eggleton [52] and Thomas [53]. 

Briefly, 100 µL of human plasma mixture was incubated with 10 µL of solution containing different polysaccharide concentrations (SP1: 17.5; 35; 70 µg/mL; SP2: 1.1; 2.19; 4.38; 8.75; 17.5; 35 µg/mL; UFH: 0.025; 0.05; 0.1; 0.2; 0.3 µg/mL) for 1 min at 37 °C. Then, 100 µL of cephalin (Biolab-Merieux AS, Rio de Janeiro, Brazil) were added. After 2 min of incubation at 37 °C, 100 µL of 25 mM CaCl_2_ was added to the mixture, and the time required for coagulation was measured through the formation time of a stable clot capable of retaining the metallic sphere present in the well with the aid of a coagulometer (KC4 Delta). 

Activity is expressed in polysaccharide micrograms per milliliter (µg/mL). The data obtained resulted in a polynomial function, according to the equation: [T/T0] = a + b_1_.[UI] + b_2_.[UI]^2^, using a standard curve based on the 6th International Standard for Unfractionated Heparin (6th UFH) (2145 units per vial), obtained from NIBSC (Potters Bar, UK), diluted in order to obtain a UFH solution with anticoagulant activity equivalent to 10 IU/mL.

### 4.9. Cell Culture

For this study, the LLC (mouse Lewis Lung Carcinoma) cell line (murine lung carcinoma, ATCC, provided by Prof. Dr. Lubor Borsig, University of Zürich) was used. Cells were grown in Dulbecco’s Modified Eagle Medium (DMEM) supplemented with 4.5 g/mL glucose (both Sigma, St. Louis, MO, USA) and 10% fetal bovine serum (Vitrocell, Waldkirch, Germany) and incubated at 37 °C in a 5% CO_2_ humid atmosphere. Cells were removed from the culture flasks and plates using a trypsin-EDTA solution (0.25% trypsin and 1 mM EDTA) (Sigma, St. Louis, MO, USA) for subsequent expansion.

### 4.10. Cell Viability Assays (MTT)

In order to assess cell viability exposed to *M. exasperatus* TP, 5 × 10^3^ cells were plated per well in 96-well plates. Then, 24 h after plating, cells were treated with vehicle (PBS) or *M. exasperatus* TP at different concentrations (0.5 µg/mL, 5 µg/mL, and 50 µg/mL) for an additional 24 h. Next, cells were incubated with tetrazolium salt (MTT, Roche, Basel, Switzerland) at 0.5 mg/mL in a cell medium for 2 h. After incubation, the cell medium was discarded, and formazan crystals were dissolved in 200 µL of dimethyl sulfoxide (DMSO, Sigma-Aldrich, St. Louis, MO, USA) per well. Absorbance at 560 nm was acquired using a SpectraMax Plus microplate reader (Molecular Devices, San Jose, CA, USA) and analyzed using Softmax Pro software 5.4.1 version (Molecular Devices, San Jose, CA, USA). Viability was calculated as a percentage of absorbance, using control cells as a reference. 

### 4.11. Clonogenic Assay

In order to evaluate LLC cells’ ability to form colonies, a clonogenic assay was performed. Regularly cultured cells were trypsinized, quantified, and plated at 500 cells/well in 6-plate wells. *M. exasperatus* TP was added to culture medium at 0.5 µg/mL, 5 µg/mL, and 50 µg/mL, or vehicle (control condition) during the low-density plating. Cells were maintained in regular culture conditions for 72 h, then the culture medium (control and TP medium) was replaced by regular medium, and cells were cultured for an additional 7 days, with medium changes every 48 h. After a total of 10 days, cells were rinsed with PBS, fixed in 100% ethanol for 1 h, and stained for 15 min with 0.4% trypan blue. Wells were imaged in a stereomicroscope and analyzed using ImageJ software for colony quantification. 

### 4.12. Wound Healing—Cell Migration—Assay

Cells were plated in 6-well plates at 1.5 × 10^5^ cells/well and cultured until 80% confluence. After reaching adequate confluence, cells were mechanically removed from the plate in a cross-shaped pattern using a sterile P1000 tip. Wells were rinsed for removal of cell debris and incubated in control (regular culture medium) or TP-medium at 0.5 µg/mL, 5 µg/mL, and 50 µg/mL for 24 h. In order to evaluate cell migration and wound closure, cells were imaged using the cross-center as a reference at 0 h and 24 h. Migration was quantified using ImageJ software and expressed as the percentage of migrated distance relative to the original wound width. 

## 5. Conclusions

This study revealed that the ascidian *Microcosmus exasperatus* is a promising source of complex and long-sulfated polysaccharides with potential therapeutic applications. *M. exasperatus* occurs in different parts of the world; therefore, its large-scale production could be developed by focusing on sustainable approaches while maintaining a high yield of glycans without great environmental impacts. We have tested the antitumor effects of *M. exasperatus* total polysaccharides; however, further fractionation is possible and scalable as well. In conclusion, we indicate that marine invertebrates’ glycans, especially those produced by ascidians, should be continuously studied as they represent an interesting alternative to mammalian glycans. 

## Figures and Tables

**Figure 1 pharmaceuticals-16-01401-f001:**
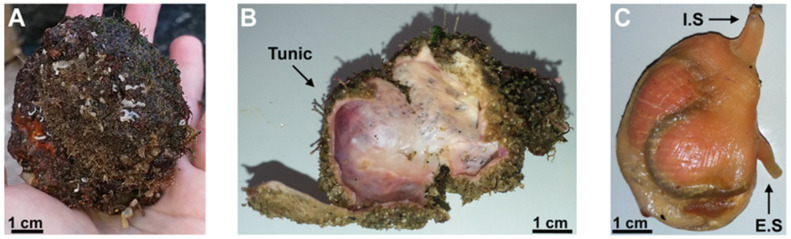
Morphology of the ascidian *Microcosmus exasperatus*. (**A**) *M. exasperatus* before dissection; (**B**) *M. exasperatus* after dissection—tunic; (**C**) *M. exasperatus* after dissection—viscera (I.S.: inhalant siphon; E.S.: exhalant siphon). Scale bar: 1 cm.

**Figure 2 pharmaceuticals-16-01401-f002:**
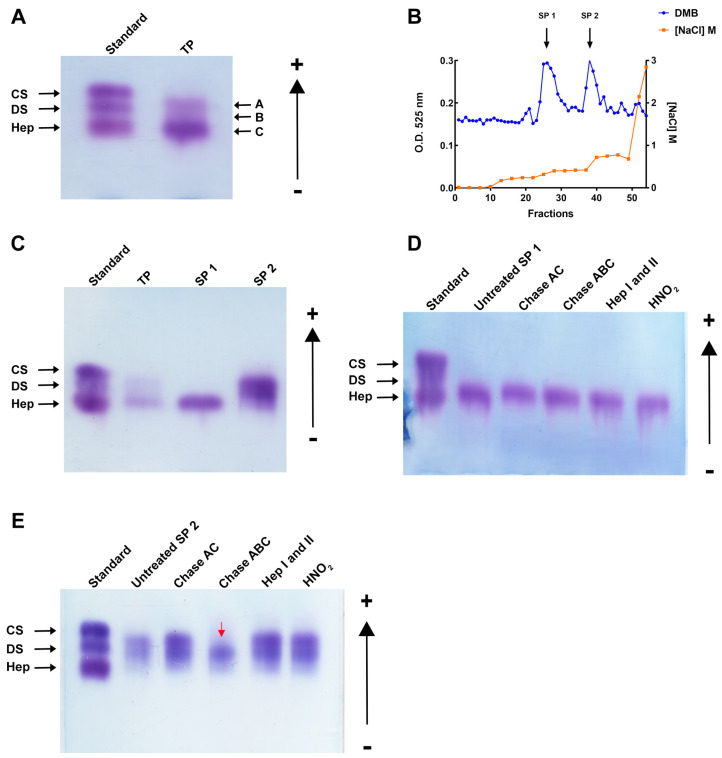
*M. exasperatus* presents three polysaccharides in its viscera. (**A**) Total polysaccharides separation by agarose gel electrophoresis-GAG standard (CS: shark chondroitin sulfate; DS: mammalian dermatan sulfate; Hep: porcine heparin), *M. exasperatus* total polysaccharides (TP), arrow A: dermatan sulfate, arrow B: heteropolysaccharide, arrow C: fucosylated-dermatan sulfate. (**B**) Ion exchange chromatography for GAG fractionation in a stepwise gradient chromatogram. Metachromatic GAG detection by DMB (blue line) and NaCl gradient (orange line). (**C**) Agarose gel of fractionated sulfated polysaccharides (SP1-fucosylated-dermatan sulfate; SP2-dermatan sulfate-containing fraction). Specific degradation treatments are needed for GAG identification. SP1 and SP2 were submitted to lyases (Chase-chondroitinase-AC and ABC; Hep-heparinases-I and II) as well as deamination with HNO_2_, (**D**) SP1 and (**E**) SP2. The red arrow shows the degraded band. GAGs standard; untreated; Chondroitinase AC; Chondroitinase ABC; Heparinases I and II mixture; HNO_2_. All gels were stained with toluidine blue.

**Figure 3 pharmaceuticals-16-01401-f003:**
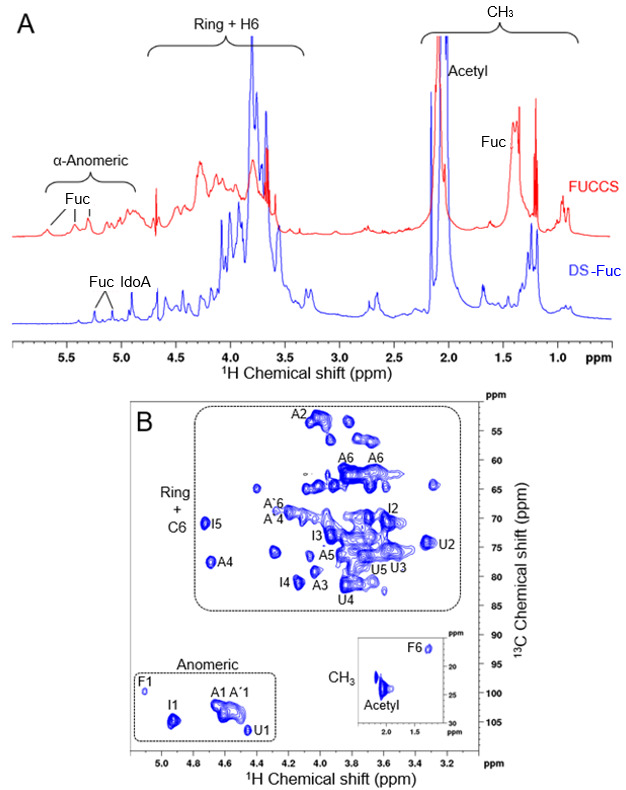
SP1 NMR spectra reveal that *M. exasperatus viscera* presents a fucosylated-dermatan sulfate. (**A**) Comparison of 1D and ^1^H NMR spectra from SP1 and fucosylated-chondroitin sulfate (FUCCS) from the sea cucumber *Ludwigothurea grisea* (in blue and red, respectively). (**B**) Two-dimensional ^1^H-^13^C HSQC experiment. Important regions, such as anomeric and ring atoms, are highlighted; the CH_3_ region is presented as a small inset. Signals are assigned as iduronic acid (I) units, 4-sulfated GalNAc (A), 6-sulfated GalNAc (A’), glucuronic acid (U), and fucose (F).

**Figure 4 pharmaceuticals-16-01401-f004:**
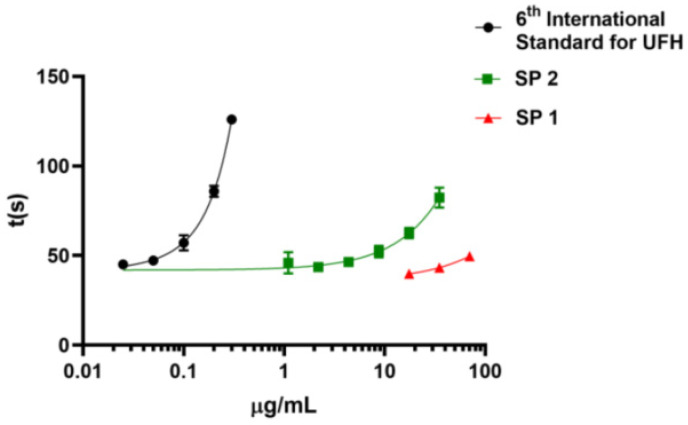
Evaluation of activated partial thromboplastin time (APTT) of *M. exasperatus* dermatan sulfate-containing fraction and UFH (unfractionated heparin) The x-axis shows the concentrations of the polysaccharides, and the y-axis shows coagulation time in seconds. The green line shows the *M. exasperatus* dermatan sulfate-containing fraction (SP2), the red line shows the fucosylated-dermatan sulfate (SP1), and the black line shows the 6th International Standard for Unfractionated Heparin. The ascidian dermatan sulfate presented, approximately, one hundred times lower activity compared to the heparin standard. *n* = 3.

**Figure 5 pharmaceuticals-16-01401-f005:**
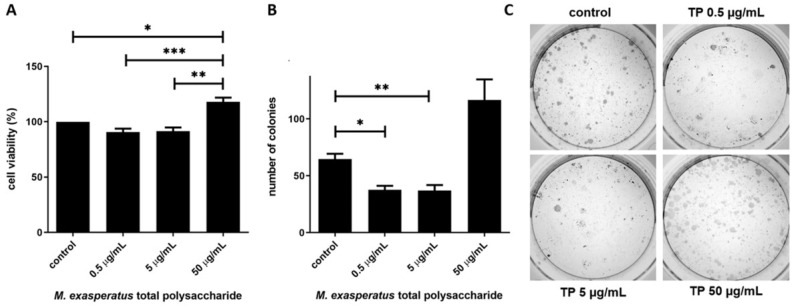
*M. exasperatus* total polysaccharides (TP) reduce tumor cell viability in a long-term, low-concentration treatment. (**A**) MTT assays and (**B**,**C**) clonogenic assays were performed in order to evaluate mouse Lewis lung carcinoma cells’ susceptibility to *M. exasperatus* TP. ANOVA with Bonferroni post-test. * *p* < 0.05, ** *p* < 0.01, and *** *p* < 0.001, *n* = 3.

**Figure 6 pharmaceuticals-16-01401-f006:**
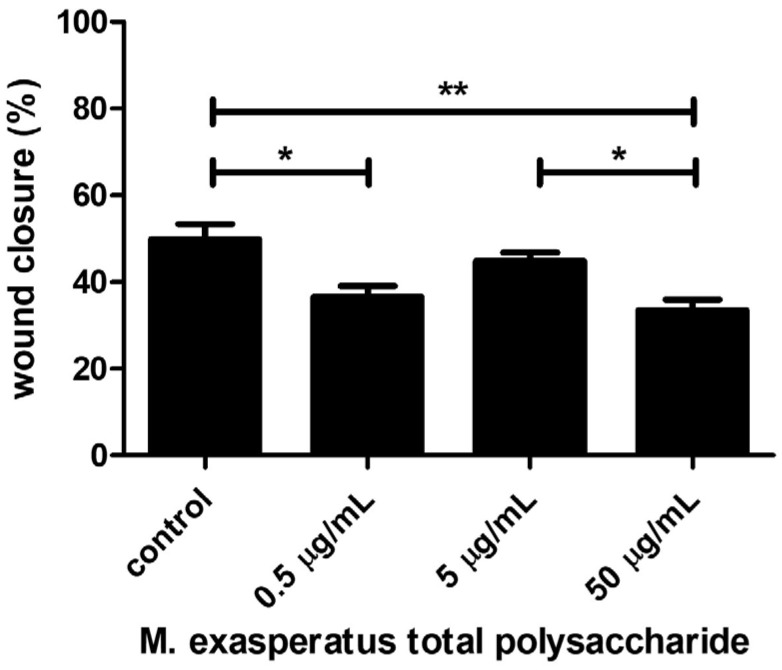
*M. exasperatus* total polysaccharides (TP) affect tumor cell migration. Mouse Lewis lung carcinoma cells were evaluated regarding their migration behavior in a wound healing/scratch assay. ANOVA with a Bonferroni posttest. * *p* < 0.05 and ** *p* < 0.01. *n* = 3.

**Table 1 pharmaceuticals-16-01401-t001:** Chemical shift comparison of SP1 and literature data.

Units	C-H1	C-H2	C-H3	C-H4	C-H5	C-H6
GalNAC-4S ^#^	4.58–103.6	4.02–54.0	4.00–78.5	4.76–79.2	3.83–77.4	3.77–63.8
GalNAC-4S ^a^	4.67/4.55–103.2	4.01–53.0	4.03–79.4	4.70–77.9	3.87–76.3	3.82/3.66–62.6
GalNAC-6S ^#^	4.58–103.6	4.02–53.4	3.98–81.7	4.20–70.3	3.97–75.1	4.24–70.1
GalNAC-6S ^a^	4.67/4.55–103.2	3.98–53.0	3.98–81.7	4.19–69.1	-	4.19–69.1
GlcA ^#^	4.46–106.5	3.37–75.2	3.57–76.2	3.77–83.8	3.66–79.5	-
GlcA ^a^	4.46–106.6	3.34–74.3	3.57–75.3	3.82–82.0	3.72–76.5	-
IdoA ^*^	4.90–102.8	3.53–71.6	3.95–73.2	4.10–82.3	4.72–72.4	-
IdoA ^a^	4.92–104.0	3.57–71.1	3.94–72.9	4.13–81.1	4.73–71.2	-

* [44], ^#^ [29], ^a^ This work, see HSQC Figure 3.

## Data Availability

Data is contained within the article.

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
