# Peer review of "Extraction, Isolation, Characterization, and Biological Activity of Sulfated Polysaccharides Present in Ascidian Viscera Microcosmus exasperatus"

_pharmaceuticals, 2023, doi:10.3390/ph16101401_

Round 1

Reviewer 1 Report

Overall. this is a good study worth publishing.  The manuscript reads like a chapter from a thesis rather than a manuscript.  There is too much information in the introduction and the beginning of the discussion reads like the introduction rather than bringing out the significance of the study.  This will be publishable after considerable editing. 

Pg2 lines 45-64                 This is an excellent introduction to ascidians for a book chapter.  It is too long for a journal article and should be condensed to one shorter paragraph containing only the essential information need by the reader.

Pg2 lines 65-67                 This sentence is too long, change “that has not been much studied so far can be found” to more straight forward English.

Pg 2 line 84                        Should “then” be now?  The sentence following it is not clear as well.

Pg 4 lines 175-240            This belongs in the methods section. The results should succinct and describe the graphs presented in this section.

Pg 4 lines 309-311            This belongs in the conclusions of the manuscript.

The language is acceptable

Author Response

We would like to acknowledge the careful work performed by reviewer 1 in analyzing our manuscript. We have modified the manuscript according to the valuable suggestions and requests and revised the English with the aid of a non-native speaker expert. We hope to have addressed all concerns with the applied changes.

In addition to changes made to the manuscript, we have included the following point-by-point reply.

Reviewer Comments:

Reviewer 1

Overall. this is a good study worth publishing.  The manuscript reads like a chapter from a thesis rather than a manuscript.  There is too much information in the introduction and the beginning of the discussion reads like the introduction rather than bringing out the significance of the study.  This will be publishable after considerable editing.

Answer:

We agree with the reviewer and have, therefore, revised our introduction to be more concise and focused, while still covering basic concepts that will allow readers from different backgrounds to understand the work.

Pg2 lines 45-64 This is an excellent introduction to ascidians for a book chapter.  It is too long for a journal article and should be condensed to one shorter paragraph containing only the essential information need by the reader.

Answer:

            We have followed the suggestion and reduced the amount of text dedicated to ascidians in the introduction.

Pg2 lines 65-67 This sentence is too long, change “that has not been much studied so far can be found” to more straight forward English.

Answer:

            We have changed the sentence portion “that has not been much studied so far” to “understudied”.

Pg 2 line 84 Should “then” be now?  The sentence following it is not clear as well.

Answer:

            We have changed the sentence from “until then” to “until now”.

Pg 4 lines 175-240 This belongs in the methods section. The results should succinct and describe the graphs presented in this section.

Answer:

            We have edited this section of results accordingly.

Pg 4 lines 309-311 This belongs in the conclusions of the manuscript.

Answer:

            We have removed this sentence from the results section according to the suggestion.

Reviewer 2 Report

The manuscript entitled” Extraction, isolation, characterization and biological activity of sulfated polysaccharides present in ascidian viscera Microcosmus exasperates” describes the protocols for sulfated polysaccharides extraction and purification targeting large-scale production and clinical applications, including cancer treatment.

Overall, the manuscript is well-written, and it could be accepted after some modifications suggested below.

Abstract: The abstract should be written concisely. The current abstract is vague.  

Discussion: The discussion is too long. It should be revised with updated references.

The author mentioned that the molecules will be applied in future preclinical studies as a soluble formula, as well as in nanoparticles, for cancer treatment. It is very promising; however, specific discussion and showing the results in this aspect are essential to establish their claim. 

Extensive editing of the English language is required

Author Response

We would like to acknowledge the careful work performed by reviewer 2 in analyzing our manuscript. We have modified the manuscript according to the valuable suggestions and requests and revised the English with the aid of a non-native speaker expert. We hope to have addressed all concerns with the applied changes.

In addition to changes made to the manuscript, we have included the following point-by-point reply.

Abstract: The abstract should be written concisely. The current abstract is vague.

Answer:

            We have thoroughly revised the abstract.

Discussion: The discussion is too long. It should be revised with updated references.

The author mentioned that the molecules will be applied in future preclinical studies as a soluble formula, as well as in nanoparticles, for cancer treatment. It is very promising; however, specific discussion and showing the results in this aspect are essential to establish their claim.

Answer:

            We have revised the discussion and updated references specially focusing on highlighting the importance of studying glycosaminoglycan-mediated antitumoral effects in different formulations, soluble and nanoparticles.

Reviewer 3 Report

This paper describes the isolation and separation of glycosaminoglycans (GAGs) from the viscera of the ascidian Microcosmos exasperatus, together with a partial structural characterization of the individual GAGs and a short study of the effects of the total polysaccharide fraction on tumor cells.

Agarose gel electrophoresis indicates three polysaccharide populations are present, and ion exchange chromatography separates two fractions; SP1, shown by NMR to be a fucosylated dermatan sulfate (DS), and SP2, a mixture of the other two polysaccharides. SP2 has anticoagulant activity and contains a DS susceptible to chondroitinase ABC.  

The effect of total M. exasperatus polysaccharide on lung cancer cells was not monotonic with concentration in any of the tests, short term viability, longer-term colony formation or cell migration. In each case the highest concentration increased cell growth/mobility, whereas lower concentrations reduced them.

Main points:

NMR analysis of SP1 does not identify the nature of linkage between fucose and the backbone chain. Is there any evidence that it is bound to the iduronate disaccharides or is it more likely to be attached at C3 of GlcA in CS-like domains of the backbone?  Is the claim on lines 412-413 that this is a fucosylated DS secure? Is SP1 just a mixture of fucosylated CS and non-anticoagulant DS?

Is the anticoagulant activity of SP2 a property of the DS fraction or of the other, unidentified polysaccharide? Does chondroitinase ABC treated SP2 retain activity? If this experiment was not done, why not?

Minor points

Lines 53-55 implies that the tunic of all ascidians is made of cellulose, but M. exasperatus on lines 77-78 has a tunic predominantly made of L-galactan – this sounds contradictory.

Lines 104-105: the statement that ‘contamination with prion proteins or even viruses remains present in this model yet’ requires support from a citation; or possibly changing ‘contamination’ to the ‘risk of contamination’, as in ref. 35.

Fig. 3 caption, line 245, should read ‘Important regions such as anomeric and ring atoms are highlighted; the CH3 region is presented as a small inset.’ The methyl part of the spectrum has not only been highlighted but also moved quite a long way.

In the Method section, section 4.9 does not mention what type of cells were used; they were presumably the lung carcinoma cell line LLC cells? How were they obtained?

English language is OK

Author Response

We would like to acknowledge the careful work performed by reviewer 3 in analyzing our manuscript. We have modified the manuscript according to the valuable suggestions and requests and revised the English with the aid of a non-native speaker expert. We hope to have addressed all concerns with the applied changes.

In addition to changes made to the manuscript, we have included the following point-by-point reply.

Main points:

NMR analysis of SP1 does not identify the nature of linkage between fucose and the backbone chain. Is there any evidence that it is bound to the iduronate disaccharides or is it more likely to be attached at C3 of GlcA in CS-like domains of the backbone?  Is the claim on lines 412-413 that this is a fucosylated DS secure? Is SP1 just a mixture of fucosylated CS and non-anticoagulant DS?

Answer:

            Unfortunately, we could not determine the specific site of fucose glycosylation. During the initial stages of NMR spectra analysis, we considered the possibility of having a mixture of polysaccharides, as suggested. However, the presence of a single, well-defined band in the agarose gel electrophoresis and a single peak in the ion-exchange chromatography indicates that it is not a combination of two different polysaccharides.

Is the anticoagulant activity of SP2 a property of the DS fraction or of the other, unidentified polysaccharide? Does chondroitinase ABC treated SP2 retain activity? If this experiment was not done, why not?

Answer:

            We have not addressed SP2 anticoagulant activity because our main focus was on antitumoral activity. Among the concentrations tested, our assays indicate that the low concentrations (0.5 and 5 µg/mL) of M. exasperatus polysaccharides are the ones with antitumoral activity. At this concentration, SP2 does not present effective anticoagulant activity.

Minor points

Lines 53-55 implies that the tunic of all ascidians is made of cellulose, but M. exasperatus on lines 77-78 has a tunic predominantly made of L-galactan – this sounds contradictory.

Answer:

            Ascidians present the extraordinary characteristics of synthesizing cellulose in addition to sulfated glycosaminoglycans. However, the first sentence (previously line 53-55) regards ascidians in general, whereas the second sentence (previously lines 77-78) concerns specific studies performed on M. exasperatus. Nevertheless, the first sentence has been removed from the manuscript due to reviewers 1 and 2 suggestions of shortening the introduction.

Lines 104-105: the statement that ‘contamination with prion proteins or even viruses remains present in this model yet’ requires support from a citation; or possibly changing ‘contamination’ to the ‘risk of contamination’, as in ref. 35.

Answer:

            The aforementioned text has been removed from the manuscript during revision, however, we agree with the reviewer that the original sentence should be changed to “risk of contamination” in case it would have remained in the manuscript.

Fig. 3 caption, line 245, should read ‘Important regions such as anomeric and ring atoms are highlighted; the CH3 region is presented as a small inset.’ The methyl part of the spectrum has not only been highlighted but also moved quite a long way.

Answer:

            We agree with the reviewer and, therefore, we have adjusted the caption for Figure 3 as suggested. It now reads, "... Important regions, such as anomeric atoms, ring are highlighted, and the CH3, region is presented as a small inset." Additionally, we have relocated the methyl region accordingly. Thank you for bringing this to our attention, and your feedback is greatly appreciated.

In the Method section, section 4.9 does not mention what type of cells were used; they were presumably the lung carcinoma cell line LLC cells? How were they obtained?

Answer:

            We apologize for the lack of adequate information regarding the cell line used in the study to evaluate antitumoral activity. Indeed, we have used LLC cells and they were originally obtained from ATCC by a collaborator and kindly donated to our group. We have edited the methods section to add this information.

Round 2

Reviewer 1 Report

Thank  you to the authors for making the appropriate changes.

Reviewer 2 Report

N/A

Reviewer 3 Report

Thanks to the authors for their clarifications.